# Effects of Postdischarge High-Protein Oral Nutritional Supplements and Resistance Training in Malnourished Surgical Patients: A Pilot Randomized Controlled Trial

**DOI:** 10.3390/nu14132599

**Published:** 2022-06-23

**Authors:** Poula Patursson, Grith Møller, Bjartur Bernhardson Thomsen, Eyðfinnur Olsen, Jann Mortensen, Guðrið Andorsdóttir, Magni Mohr, Jens Rikardt Andersen

**Affiliations:** 1Department of Surgery, National Hospital of the Faroe Islands, J.C. Svabosgøta 41–49, 100 Torshavn, Faroe Islands; eydol@ls.fo; 2Department of Nutrition, Exercise and Sports, Faculty of Science, University of Copenhagen, Rolighedsvej 26, 1958 Frederiksberg, Denmark; gmp@nexs.ku.dk (G.M.); jra@nexs.ku.dk (J.R.A.); 3Department of Occupational and Physiotherapy, National Hospital of the Faroe Islands, J.C. Svabosgøta 41–49, 100 Torshavn, Faroe Islands; bjartur1988@hotmail.com; 4Department of Clinical Physiology, Nuclear Medicine & PET, Centre of Diagnostic Investigation, Copenhagen University Hospital, Blegdamsvej 9, 2100 Copenhagen, Denmark; jann.mortensen@regionh.dk; 5The Genetic Biobank of the Faroes, J.C. Svabosgøta 43, 100 Torshavn, Faroe Islands; gudrid@biobank.gov.fo; 6Centre of Health Sciences, Faculty of Health, University of the Faroe Islands, Vestara Bryggja 15, 100 Torshavn, Faroe Islands; magnim@setur.fo; 7Department of Sports Science and Clinical Biomechanics, SDU Sport and Health Sciences Cluster (SHSC), University of Southern Denmark, Campusvej 55, 5230 Odense, Denmark

**Keywords:** malnutrition, surgery, postdischarge, resistance training, oral nutritional supplements

## Abstract

The presence of malnutrition is increasingly becoming a postdischarge problem in surgical patients. We aimed to investigate whether oral nutritional supplements combined with resistance training could minimize skeletal muscle atrophy in surgical patients after discharge. This randomized controlled study was conducted at the Department of Surgery, National Hospital of Faroe Islands from 2018 to 2020. A total of 45 patients aged 37–74 years participated and were allocated to one of three groups: diet (DI; *n* = 13), exercise and diet (EX + DI; *n* = 16), or control (CON; *n* = 16). The intervention period lasted 8 weeks. The intervention groups received individual dietary counselling and a protein-rich oral nutritional supplement twice a day containing 22 g of protein/day. Patients in the EX + DI group were assigned to resistance training sessions. Patients in the CON group received standard care. The primary outcome was change in lean body mass (LBM). Secondary outcomes were change in body weight, handgrip strength, quality of life, surgery-related side effects, energy and protein intake, length of stay and one-year mortality. To estimate within-group changes, linear mixed models including group–time interactions as fixed effects and patients as random effects were fitted. Within-group change in LBM was 233, 813 and 78 g in the DI, EX + DI and CON groups, respectively, with no significant between-group difference (*p* > 0.05). Pain score declined more (*p* = 0.04) in the EX + DI group compared with the CON group. Body weight, handgrip strength, quality of life and surgery-related side effects did not differ between groups. At the end of study, mean cumulative weight change in the DI and EX + DI groups was 0.4% and 1.6%, respectively, whereas the CON group experienced a weight loss of −0.6%. No significant difference in primary outcome between groups was noted. However, our results indicate some benefits from exercise and nutrition for malnourished surgical patients.

## 1. Introduction

Around 50% of surgical patients are malnourished during hospitalization [1], and nutritional status declines further during admission and after discharge [1,2,3,4]. Hospitalization periods are normally short, and the real rehabilitation starts at home. Thus, the presence of malnutrition is increasingly relocating to the postdischarge setting.

Postoperative immobilization and loss of appetite are common in surgical patients [4], and hormonal and metabolic changes, i.e., the surgical stress response, frequently occur following surgery [5,6], leading to deterioration in protein synthesis and accelerated protein catabolism [5,7,8]. Skeletal muscle mass is associated with muscle strength and functional abilities [9,10]. Hence, loss of muscle mass affects patients’ general condition, quality of life (QOL) and physical performance.

In recent years, perioperative malnutrition has been acknowledged by the surgical community, and the importance of early postoperative feeding is being recognized as a major factor in reducing surgical complications [8,11,12,13,14,15] and enhancing recovery after surgery [16,17,18]. There are indications that protein supplementation could reverse the negative nitrogen balance in the immediate postoperative stage after gastrointestinal surgery [19], and hence affects weight and nutritional status. However, postoperative protein supplementation is not routinely used, even when malnutrition is identified [12].

Randomized controlled trials have shown that nutritional counselling and supplementation may improve nutritional status in malnourished patients, even following discharge [4]. Beattie and colleagues [20] investigated the effect of oral nutritional supplements (ONS) in surgical patients for 10 weeks and found that both groups continued to lose weight postdischarge, but after two weeks the intervention group started gaining weight, resulting in a significant lower net weight loss. Additionally, it is well known that dietary protein and resistance training stimulate skeletal muscle hypertrophy [21]. The intake of protein after resistance training increases the plasma amino acids concentration, which results in the activation of signaling molecules leading to increased protein synthesis and concomitant skeletal muscle hypertrophy [22]. Therefore, the combination of resistance training and protein supplementation may be beneficial for patients postsurgery, since this period is associated with severe weight loss and insufficient protein intake [1,2,3].

Miller and colleagues examined the effects of nutritional supplementation and resistance training in elderly following a lower-limb fracture and showed a beneficial effect on weight from combined nutritional and exercise training interventions [23]. However, the study did not examine the effect on lean body mass and was therefore unable to explain the observed changes in body composition.

To the best of our knowledge, no studies have investigated the effects of resistance training combined with protein supplementation on body composition using dual-energy X-ray absorptiometry (DXA) to monitor changes in LBM after discharge in surgical patients.

The aim of the present study was to investigate whether oral nutritional supplements (ONS) combined with resistance training could minimize skeletal muscle atrophy often observed in surgical patients after discharge.

Based on the existing evidence, we hypothesize that combined nutritional support and resistance training prevent loss of body weight and LBM in malnourished surgical patients following discharge compared with oral nutritional support alone or standard care.

## 2. Materials and Methods

This study was a prospective three-arm block pilot randomized controlled trial. Eligible patients were consecutively enrolled in the study between April 2018 and January 2020 at two surgical wards at the National Hospital of the Faroe Islands, Torshavn, Faroe Islands.

Eligible patients were randomly assigned to one of three groups: diet (DI), exercise and diet (EX + DI), or control (CON) using stratified block randomization according to gender and whether the surgery was open or minimally invasive. The patients were randomized using opaque, sealed envelopes with 12 blocks each consisting of six envelopes. The investigators performed data collection and analysis; thus, blinding to the treatment allocation was not possible.

### 2.1. Inclusion and Exclusion Criteria

The inclusion criteria were: postoperative patients (≥18 years of age); recent elective or acute gastrointestinal surgery, breast cancer surgery, gynecological surgery, acute femoral neck fracture, leg amputation and elective hip and knee alloplastic; presence of nutritional risk at time of assessment according to NRS 2002 [24]; planned to be discharged from hospital within 1–2 days after surgery.

Patients were excluded from the study if they had impaired kidney function defined as p-creatinine > 250 µmol/L or dialysis treatment; strict training restrictions (e.g., after back surgery); minor orthopedic surgeries such as arm, hand and foot surgeries; exclusive enteral or parenteral nutrition; same-day surgery; terminal illness; language or cognitive impairment or admittance to intensive care.

### 2.2. Intervention

The intervention began at the day of discharge and lasted eight weeks. The two intervention groups received two protein-rich ONS daily (Nestle Komplett Näring, 400 mL/day, and 22 g protein/day). Furthermore, patients in the DI and EX + DI groups received individual dietary counselling by a registered dietitian on high-protein noncommercial foods. The aim of the individual counselling was to meet patients’ nutritional requirements and prevent further deterioration of nutritional status.

The EX + DI group completed 16 supervised and 8 unsupervised training sessions over the 8-week intervention period. Two supervised training sessions were offered per week either at the National Hospital Physiotherapy ward or at a local physiotherapist. The 4 exercises lasted 35 min in total and were proceeded by a 10 min warm-up, the exercises for the upper body were: row with an elastic band and shoulder flexion with an elastic band. For the lower body the exercises were: sit to stand; if the participant was unable to stand, they performed leg extensions with an elastic band as resistance. The participant also carried out heel raises. The first week (week 1) aimed to familiarize the participants with the training program. The next three weeks (week 2–4) participants performed two sets of 15 repetitions of all exercises. This corresponded to 45–60% of their 1 repetition maximum (RM). The aim was to reach contraction failure (muscular fatigue) at a relative load zone of 12–15 RM in each set. During the next 4 weeks (week 4–8), the participants were asked to perform three sets of 12 RM of each exercise. This corresponded to 60–70% of 1 RM and was performed until contraction failure at a relative load zone of 8–12 RM in each set. A 2 min pause was allowed between sets. The correct level of each exercise was chosen according to the progression models by the supervising physiotherapist. Each set of each exercise was considered unique and determined whether the patient stayed on the same level or either progressed or regressed. The upper-body exercises were progressed by adding resistance to the elastic band. The lower-body exercises were progressed by adding external load.

### 2.3. Control Group

Control patients received standard nutritional care including appointed follow-ups as provided in the hospital or at home. They were advised to eat as usual and ONS or physical training was only provided if prescribed by the doctor at discharge.

### 2.4. Outcomes

The primary outcome was change in LBM.

Secondary outcomes were mean energy and protein intake, change in BW, handgrip strength (HGS), QOL, surgery-related side effects, complications and activities of daily living. BW was measured at baseline and every second week during the intervention period, whereas all other outcomes were assessed at baseline and the end of the study.

Surgery-related side effects were assessed using a questionnaire at baseline and at the end of the study. The questionnaire was constructed by the research team using typical adverse surgical effects in malnourished patients. The questions comprised the categories pain, severeness of pain and intake of pain killers, diarrhea, nausea, emesis, loss of appetite, flatulence and mobility after surgery [25,26]. In addition, total length of stay in hospital was registered at six months post surgery and post inclusion to the study. One-year all-cause mortality was registered.

### 2.5. Measurements

Whole-body composition was assessed at baseline and post intervention using DXA (Norland XR-800, Illuminatus software, Norland Corporation, Oslo, Norway). Measures of body composition included total and regional body fat, lean tissue and bone mineral density (BMD). The effective radiation dose was <0.2 mSv per scan.

BW was measured with patients wearing light clothes on a chair scale (Seca digital chair scale, Seca, CA, USA) on the surgical ward at baseline and post intervention. After discharge, every second week patients measured their BW at home using a household scale. Height was measured in centimeters at baseline using a stadiometer.

HGS was measured on the patients’ dominant hand in an upright, sitting position with the elbow at 90 degrees using a Camry digital hand dynamometer (Camry electronic hand dynamometer EH101, Zhongshan Camry Electronic Co., China). We standardized the measurements of HGS by using the same position for repeated measurements after we demonstrated the technique once to every patient. We encouraged them to squeeze the hand dynamometer with maximum strength for three seconds and repeat it twice. The highest measurement was used in the data analysis.

QOL was assessed using the validated Short Form-12 (SF-12, second version, Medical Outcomes Trust 2006). Patients were asked to answer 12 questions concerning eight dimensions of health. For data analyses, only changes in QOL scores were compared between groups, not the actual scores.

Functional mobility and walking performance were examined using a De Morton Mobility Index (DEMMI) test. The DEMMI test measures mobility in 15 items in the spectrum from bed-bound patients to fully independent patients [27]. Lower-body muscle strength was measured using a 30 s Chair Stand Test. Patients were instructed to rise from a regular chair and sit back down again as often as they could for 30 s [28]. The number of rises and sits was recorded and compared with the average age and gender-specific scores. A score below 8 was considered abnormal for all ages. The change in score during intervention was compared between groups.

Blood samples were collected at baseline and post intervention. Analyses included micronutrients (Cobalamin, Ferritin, Iron, Transferrin, Hemoglobin, Transferrin iron binding capacity, Magnesium, Phosphate, Folate, Zinc and 25-Hydroxy-Vitamin D) to ensure that patients did not suffer from micronutrient deficiencies. In addition, inflammation markers (C-Reactive Protein, Leucocyte count and p-ferritin) were measured to estimate metabolic stress level that potentially may influence the interpretation of micronutrient status due to acute phase reactant effects.

Nutritional requirements were estimated based on the Harris and Benedict equation to determine basic metabolic rate (BMR) [29] and then multiplied with an estimated activity factor and, when needed, a stress factor [30]. The stress factor was set at 1.3 for patients with open surgery and 1.2 for patients with less invasive surgery. The activity factor was assessed individually.

Protein requirements were estimated as 1.5 g/kg/d because of the expected stress metabolism [30].

Dietary intake was assessed using 24 h food records at baseline and at the end of the study. In the cases where patients did not manage to record their food intake, 24 h diet recall interviews were performed by a trained dietitian. Dietary intake was calculated using a nutrition analysis software application (Vitakost©, Vitakost ApS, Kolding, Denmark). A sufficient nutritional intake was defined as minimum 75% of the estimated nutritional requirements [31].

### 2.6. Compliance

All intervention patients kept an ONS diary showing amount and time of ONS throughout the intervention period. Adherence to dietary recommendations and ONS was evaluated by regular phone interviews every two weeks. Furthermore, patients in the EX + DI group kept a training diary, and additionally, training attendance was evaluated by checking medical charts.

### 2.7. Statistical Analysis

To estimate within-group changes from baseline and between-group differences in changes, linear mixed models including group–time interactions as fixed effects and patients as random effects were fitted. Age was also included as a fixed-effect covariate. Intention-to-treat analyses and per protocol analyses for patients with at least 75% compliance were carried out. Simple linear regression analyses were fitted to estimate the effect of compliance on changes in outcome (ANOVA). All statistical analyses were carried out using R [32]. A significance level of 0.05 was used.

## 3. Results

Totally, 57 patients were eligible for inclusion during the study time from May 2018 to April 2020 (Figure 1).

Ten patients declined to participate in this study due to general health and wellbeing (*n* = 7), very old age (*n* = 1), living remotely and not being able to attend training classes (*n* = 1), predementia (*n* = 1) and no reason given (*n* = 2). In total, 45 patients were randomized, and 40 patients (89%) completed the study. Five patients (11%) dropped out, four in the CON and one in the DI group.

Baseline characteristics are summarized in Table 1.

Patients in the EX + DI group were younger than those in the CON and DI groups. All patients were moderately undernourished (mean (SD) NRS 2002 nutritional score of 1.25 (0.9)) and, overall, had a moderate severity of disease score (mean (SD) severity of disease score of 1.64 (0.5)).

### 3.1. Intention to Treat Analysis

The EX + DI group had a change in LBM of median (95% CI) 813 (−873, 2499) g during the intervention period (*NS*), whereas the changes in the DI and CON groups were 233 (−1712, 2178) g and 78 (−1948, 2105) g, respectively (Table 2). The differences between groups were not significant.

#### 3.1.1. Weight Change during the Intervention Period

At the end of the study, mean cumulative weight change for the combined EX + DI intervention was 1.6% compared with 0.4% in the DI group and −0.6% in the CON group (Figure 2). During the eight-week intervention period, the EX + DI group did not experience weight loss, whereas the DI group continued to lose weight until week 2, whereafter the weight stabilized and started to increase from week 4. However, in the control group, weight loss continued until week 6 (Figure 2).

#### 3.1.2. Energy and Protein Intake

The EX + DI group increased their energy intake significantly (*p* = 0.01) during intervention (Table 2). There was a greater change in energy intake in the DI group (*p* = 0.04) and EX + DI group (*p* = 0.01) compared with the CON group. The DI and EX + DI groups had the greatest increase in protein intake but not significantly different from the CON group (*p* > 0.05).

At the end of the intervention, mean (SD) total energy intake in the DI, EX + DI and CON groups was 10,300 (3820), 10,500 (2780) and 7280 (2550) kJ/d, respectively. Protein intake in the DI, EX + DI and CON groups was 92.4 (33.5), 92.5 (27.3) and 73.2 (26.4) g/d, respectively, equivalent to 107 (40.6), 121 (42.3) and 82.0 (27.1) percent of energy requirements and 89.4 (34.3), 99.4 (31.7) and 76.7 (27.4) percent of protein requirements in the DI, EX + DI and CON groups, respectively.

Post intervention, the number (%) of patients reaching 75% of their minimum energy requirements was 10 (83%), 10 (91%) and 6 (67%) in the DI, EX + DI and CON groups, respectively. The number (%) of patients achieving 75% of their minimum protein requirements was 7 (58%), 9 (82%) and 4 (44%) in the DI, EX + DI and CON groups, respectively.

#### 3.1.3. Other Outcomes

There was a difference in change in pain scores between the EX + DI and CON groups (*p* = 0.04), with the EX + DI group experiencing the greatest decline in pain (Table 2). Handgrip strength did not differ between groups.

All three groups had significant increases in activities of daily living (DEMMI scores) (*p* < 0.01) and sit-to-stand scores (*p* = 0.02, *p* < 0.01, *p* < 0.01) and reductions in low self-reported food intake (*p* < 0.01) and use of painkillers (*p* < 0.01) (Table 2). However, the changes between groups were not significant.

The total length of stay in hospital 6 months postsurgery was median (IQR) 10.0 [9.0, 15.3], 9.00 [4.8, 16.3] and 5.50 [4.0, 14.5] days in the DI, EX + DI and CON groups, respectively. The total length of stay post-inclusion to the study was median (IQR) 6.0 [3.5, 110.5], 4.00 [1.0, 8.8] and 1.00 [1.0, 5.8] days in the DI, EX + DI and CON groups, respectively. One-year all-cause mortality was 0 in all groups.

Six of twelve (50%), five of sixteen (31%) and four of twelve (33%) patients in the DI, EX + DI and CON groups, respectively, experienced postoperative complications. Six-month readmission rates were 33%, 38% and 17% in the DI, EX + DI and CON groups, respectively. Four patients in the CON group were dropouts, and no data on readmission was retrieved.

### 3.2. Per Protocol Analyses

#### 3.2.1. Training Compliance

In total, 10 of 16 (63%) patients in the EX + DI group had over 75% adherence to the training protocol. Mean (±SD) attendance was 2.7 (1.3) sessions per week for 8 weeks corresponding to 88% (34) compliance. During the intervention, attendance varied from 90% during the first week, decreased to 83% in week three, and increased again to 90 % in week eight.

Patients in the EX + DI group with high training compliance (attendance >75% of training sessions) increased their LBM by 1709 (−491, 3908) g and their body weight by 2.9 (0.3, 5.4) kg during the eight-week intervention period. The change in the EX + DI group was not significantly different from the changes seen for patients in the DI or CON groups (Table 3).

#### 3.2.2. ONS Compliance

Totally, 2 of 12 (17%) DI patients and 7 of 16 (44%) EX + DI patients consumed over 75% of the prescribed dose of ONS. In total, mean (SD) ONS compliance was 56% (34). ONS compliance in DI patients was 48% and in EX + DI patients 61% (Table 4).

No correlations were observed between change in LBM and total energy intake or total protein intake.

DI patients with high ONS compliance (>75% ONS) increased their LBM by 2.1 (−0.4, 5.6) kg and their body weight by 1.4 (−4.2, 6.9) kg during the eight-week intervention period. Patients in the EX + DI group with high ONS compliance (>75% ONS) increased their LBM by 1.9 (0.0, 3.7) kg and their body weight by 2.0 (−0.9, 5.0) kg during the eight-week intervention period (Table 4). The change from pre to post was not significantly different from the CON group.

## 4. Discussion

This study is the first to investigate the effect of combined nutritional and resistance intervention in malnourished postsurgical patients.

We found no statistical differences in the change in LBM between intervention groups. The percentage of LBM change in the EX + DI group was three to sevenfold higher than in the DI and CON groups, respectively. The results may be too small to detect a significant difference between groups due to a type II statistical error.

Physical activity is a major factor in maintaining or enhancing LBM and increasing muscle strength. Accordingly, in the current study, the EX + DI group increased their LBM sevenfold compared with the CON group and increased their HGS, suggesting that exercise may increase LBM and HGS even in frail patient populations. The results are not statistically significant, perhaps due to a type II statistical error. Internationally, no studies have investigated the effects of postdischarge resistance training combined with protein supplementation on body composition in surgical patients, although some studies have applied weight loss as an endpoint [23]. Interestingly, trials in the field of clinical nutrition do not usually include physical exercise interventions [4,33].

Early mobilization is a crucial component in the field of enhanced recovery after surgery (ERAS), and exercise is generally recommended to prevent muscle atrophy and increase recovery [16]. Preoperative physical fitness is well-known to predict postoperative outcomes in general and to predict inhospital morbidity in patients undergoing major elective colorectal surgery [34].

In cancer patients, neoadjuvant and adjuvant chemotherapy reduce physical fitness and are often debilitating. Recent trials have demonstrated that preoperative training programs may reverse the side effects of chemotherapy and increase patients’ fitness levels [35].

Most surgical patients experience pain postoperatively, and even though exercise is a well-known treatment for chronic pain [36], we were surprised to find significant effects from exercise on pain in our patient group. The significantly different pain reduction in the EX + DI group suggests that postdischarge resistance training could be effective in reducing surgery-related pain. To the best of our knowledge, no studies have investigated the possible effects of exercise on surgery-related pain.

Even though we found no significant differences in weight change between groups, the EX + DI group experienced the greatest weight change. Furthermore, we observed that the pattern of weight change after discharge was different between groups, as patients in the EX-DI group managed to maintain their bodyweight during intervention, while the other two groups experienced weight loss during the first weeks. Thus, this study shows that it is possible to achieve weight maintenance in this frail patient group when combining nutritional supplements with resistance training.

Furthermore, our study demonstrated that the EX + DI group increased their energy intake more than the two other groups. Overall, patients in both intervention groups had higher coverage of their energy requirements during intervention, whereas the control group reduced their coverage of their energy requirements. A recent review suggested that exercise and physical activity promote appetite in older adults [37]. Thus, combining nutritional supplements with exercise to malnourished surgical patients may increase the percentage of patients reaching 75% of their estimated energy and protein requirements, which is considered clinically relevant.

### Strengths and Limitations

This study was underpowered to detect the expected differences between groups because of the COVID-19 pandemic restrictions of access to patients, which is a major limitation. Not only patients changed their lifestyle during the COVID-19 pandemic period. The patient population was very small, and the inclusion criteria regarding both surgery and malnutrition made some patients not eligible. Ten patients declined to participate. A short prevalence study revealed around 40% of admitted surgical patients being at risk of malnutrition, which explains why not so many patients were eligible. Furthermore, many patients are treated conservatively, meaning they do not receive surgery, and hence are not eligible.

We are confident that the program improved nutritional intake as we had a control group. However, compliance to the resistance training program is much more complicated to evaluate, as we did not have a control group, and the supporting efforts may have been insufficient due to limited possibilities for physical contact with the patients.

Potential confounders may have impacted several outcomes, such as adherence to the intervention program and the potential interaction with postoperative complications.

The randomized controlled design increases the strength of the present pilot study. The reason for choosing a three-armed design instead of a two-armed design was to investigate whether the previous gold standard treatment of only oral nutritional supplements for malnourished patients could be improved by adding resistance training as an extra treatment intervention. We were interested to assess if diet combined with exercise should be the new first-line treatment.

The low dropout rate further increases the strength of the primary endpoint results. The use of food records is a validated method for estimating energy and protein intake but also is cumbersome to the patient. Thus, we chose 24 h food records. The main advantage of dietary food records is that they are a prospective method that does not rely on the respondent’s memory. However, the short period is a limitation.

We used standardized protocols for measuring HGS with patients always sitting in the upright position, which is a strength. Assessments of activities of daily living such as the DEMMI test and the sit-to-stand test were performed by a trained physiotherapist, which improves the quality of the procedures. Body composition assessment was made using DXA, which is generally considered superior to Bioelectrical Impedance Analysis at the individual level [38].

If this novel approach combining ONS and exercise for malnourished surgical patients after discharge from hospital is to be implemented, economic implications should be considered, which we did not conduct.

Implementing supervised exercise for all malnourished patients after surgery would increase costs, which might limit the implementation of our approach. However, malnutrition has major economic consequences and detrimental effects on patients’ quality of life and clinical outcome, and hence, our approach could possibly constitute a relatively low-cost intervention for preventing and treating malnutrition in surgical patients. Future studies should include the aspect of economics.

We argue that body weight and nutritional status should be monitored during the postoperative period and the use of oral nutritional supplements and exercise, as carried out in the present study, should be considered in malnourished patients who have undergone surgery.

## 5. Conclusions

In conclusion, the intervention had a positive impact on body weight and LBM, especially in the EX + DI group, indicating that exercise combined with nutritional supplements may be a simple strategy for reducing weight loss and muscle atrophy in malnourished surgical patients. Furthermore, the intervention seemed to have some impact on relevant outcomes such as surgery-related pain, energy and protein intake, QOL and physical function. The results support the concept of integration of nutrition and resistance training as an essential part of postoperative care, but need to be tested with much better statistical power in a time without pandemic restrictions.

## Figures and Tables

**Figure 1 nutrients-14-02599-f001:**
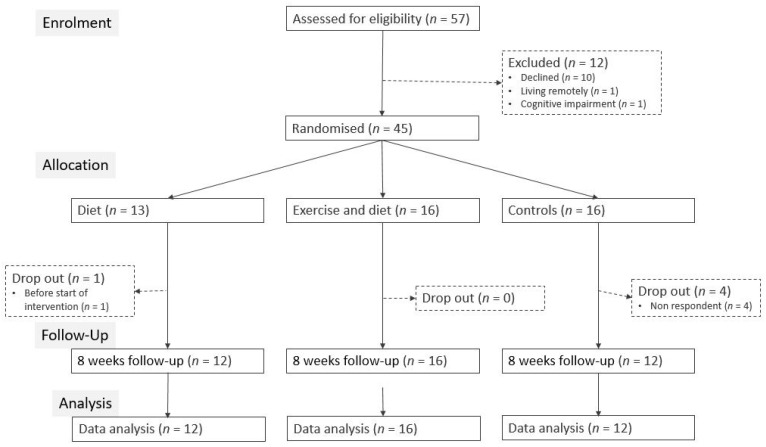
Flow diagram showing enrolment of postdischarge surgical patients and number of patients included in the analysis.

**Figure 2 nutrients-14-02599-f002:**
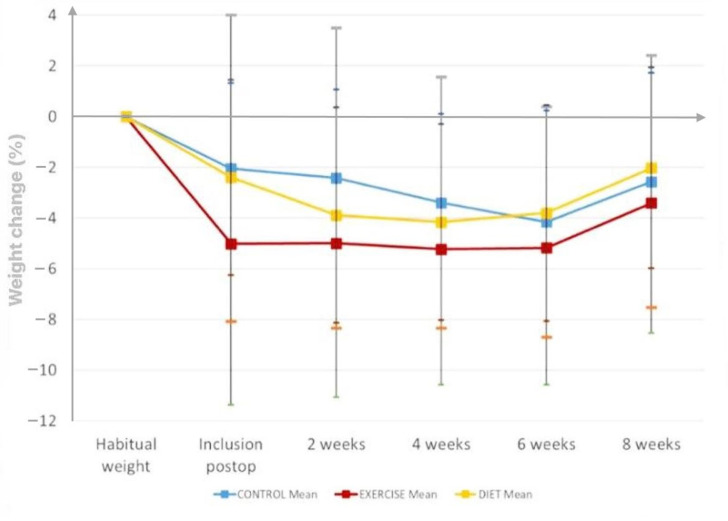
Cumulative weight change (%) in postdischarge surgical patients (mean ± SD).

**Table 1 nutrients-14-02599-t001:** Baseline characteristics of postdischarge surgical patients collected prior to discharge before randomization.

	Diet (*n =* 13)	Exercise + Diet (*n =* 16)	Control (*n =* 16)
**Sex**			
Female	7 (53.8%)	7 (43.8%)	8 (50.0%)
Male	6 (46.2%)	9 (56.2%)	8 (50.0%)
**Age**	68.0 [56.0, 74.0]	59.0 [37.3, 73.3]	66.5 [54.8, 70.3]
**Postoperative days at inclusion**	4.0 [3.0, 8.0]	6.50 [3.0, 9.0]	3.00 [3.0, 6.5]
**Surgery**			
Minimal	0 (0%)	5 (31.2%)	3 (18.8%)
Open	13 (100%)	11 (68.8%)	13 (81.2%)
**Specialty**			
Gastroenterology	9 (69.2%)	8 (50.0%)	5 (31.2%)
Gynaecology	0 (0%)	0 (0%)	1 (6.2%)
Orthopaedics	4 (30.8%)	8 (50.0%)	9 (56.2%)
Urology	0 (0%)	0 (0%)	1 (6.2%)
**Habitual bodyweight (kg)**	89.1 (10.6)	81.5 (21.5)	77.8 (18.2)
**Bodyweight (kg)**	86.7 (9.02)	77.2 (20.6)	76.3 (18.3)
**BMI**	29.6 [28.0, 32.0]	24.4 [23.6, 28.1]	25.6 [22.7, 30.8]
**Weight loss from habitual weight (%)**	−2.35 (3.70)	−4.37 (5.41)	−1.53 (4.40)
**Lean Body mass (kg)**	52.3 (7.8)	51.7 (13.9)	51.0 (11.9)
**Fat mass (kg)**	35.7 (12.1)	25.4 (13.2)	25.7 (13.5)
**Handgrip strength**	32.6 (9.72)	35.0 (13.1)	34.1 (15.3)
**Nutritional Risk Score (NRS)**			
Total NRS ≥ 3 (*n* (%))	12 (92)	14 (88)	14 (88)
NRS nutritional score (0–3)	1.1 (1.0)	1.1 (0.7)	1.5 (0.9)
NRS severity of disease (0–3)	1.9 (0.3)	1.6 (0.5)	1.4 (0.5)
Total NRS score (0–7)	3.4 (1.1)	3.1 (0.6)	3.3 (1.2)
**Energy intake**			
Energy intake (kJ/d)	8130 (3020)	7590 (2600)	8880 (2430)
Energy intake (% of requirements)	81 (30)	83 (29)	98 (35)
Energy < 75% requirements (*n* (%))	2 (15)	5 (31)	4 (25)
**Protein intake**			
Protein intake (g/d)	73.8 (30.6)	72.3 (33.4)	82.8 (19.7)
Protein intake (g/kg/d)	0.9 (0.36)	1.1 (0.64)	1.1 (0.49)
Protein intake (% of requirements)	69 (28)	73 (30)	85 (26)
Protein < 75% requirements (*n* (%))	2 (15)	6 (38)	4 (25)

Data presented as mean (SD) or median (IQR). Abbreviations: BMI: body mass index; *n*: number of patients.

**Table 2 nutrients-14-02599-t002:** Change in primary and secondary outcomes during the eight-week intervention period.

Change within Groups							Difference between Groups					
	Diet (*n* = 12)	*p*	Exercise + Diet (*n* = 16)	*p*	Control (*n* = 12)	*p*	Exercise + Diet vs. Diet	*p*	Diet vs. Control	*p*	Exercise + Diet vs. Control	*p*
LBM (kg)	0.2 (−1.7, 2.2)	0.81	0.8 (−0.9, 2.5)	0.36	0.1 (−1.9, 2.1)	0.93	0.6 (−2.0, 3.2)	0.66	0.2 (−2.7, 3.0)	0.91	0.7 (−1.9, 3.0)	0.59
LBM %	0.4 (−3.8, 4.6)	NS	1.5 (−2.1, 5.2)	NS	0.2 (−4.2, 4.6)	NS						
Weight (kg)	0.5 (−1.8, 2.8)	0.68	1.4 (−0.6, 3.4)	0.17	0.4 (−1.9, 2.7)	0.75	0.9 (−2.1, 4.0)	0.56	0.1 (−3.1, 3.4)	0.95	1.0 (−2.0, 4.1)	0.51
Weight (%)	0.5 (−2.7, 3.7)	NS	2.0 (−0.8, 4.8)	NS	0.7 (−2.5, 3.9)	NS						
Handgrip strength (kg)	1.2 (−1.5, 3.8)	0.40	2.1 (−0.2, 4.5)	0.08	1.6 (−1.1, 4.4)	0.25	1.0 (−2.6, 4.6)	0.59	−0.5 (−4.3, 3.4)	0.81	0.5 (−3.1, 4.2)	0.78
Fat mass (kg)	−0.4 (−1.8, 1.0)	0.56	−0.1 (−1.3, 1.2)	0.91	0.4 (−1.1, 1.8)	0.64	0.4 (−1.5, 2.2)	0.71	−0.8 (−2.8, 1.3)	0.46	−0.4 (−2.3, 1.5)	0.67
Energy intake (kJ/d)	2311 (−177, 4798)	0.07	2818 (−669, 4968)	0.01 *	−1402 (−3808, 1004)	0.25	507 (−2780, 3795)	0.76	3713 (252, 7174)	0.04 *	4220 (994, 7447)	0.01 *
Energy intake (% of requirements)	25 (−7, 58)	0.13	38 (9, 66)	0.01 *	−16 (−48, 15)	0.31	12 (−31, 56)	0.58	42 (−4, 87)	0.07	54 (11, 97)	0.01 *
Protein intake (g/d)	21.0 (−2.9, 44.8)	0.09	20.1 (−0.5, 40.6)	0.06	−8.6(−31.7,14.5)	0.47	00.9 (−32.4, 30.6)	0.96	29.5 (−3.7, 62.7)	0.08	28.6 (−2.3, 59.5)	0.07
Protein intake (% of requirements)	20 (−8, 48)	0.17	26 (1, 51)	0.04 *	−9 (−36, 18)	0.53	6 (−31, 44)	0.74	29 (−11, 68)	0.15	35 (−2, 72)	0.06
QOL:SF-12-1, general health, scale	−0.2 (−0.7, 0.3)	0.47	−0.5 (−1.0, 0.0)	0.03 *	−0.9 (−1.4, −0.3)	<0.01 *	−0.3 (−1.0, 0.4)	0.40	0.7 (−0.1, 1.4)	0.08	0.4 (−0.3, 1.1)	0.31
QOL:SF-12-8, pain interference, scale	−0.3 (−1.2, 0.5)	0.43	−1.4 (−2.1, −0.6)	<0.01 *	−1.0 (−1.9, −0.2)	0.01 *	−1.0 (−2.2, 0.1)	0.07	0.7 (−0.5, 1.9)	0.26	−0.3 (−1.5, 0.8)	0.50
Sit-to-stand, times	3.2 (0.5, 5.9)	0.02 *	6.0 (3.5, 8.4)	<0.01 *	6.9 (4.1, 9.7)	<0.01 *	2.8 (−0.9, 6.4)	0.14	−3.7 (−7.6, 0.2)	0.06	−1.0 (−4.6, 2.7)	0.61
DEMMI, score	17 (6, 28)	<0.01 *	18 (8, 28)	<0.01 *	15 (4, 27)	<0.01 *	1 (−14, 16)	0.88	2 (−14, 18)	0.82	3 (−12, 18)	0.71
Surgery-related side effect: pain, scale (1 no pain, 6 very strong pain)	−1.0 (−1.9, 0.0)	0.05 *	−1.5 (−2.4, −0.7)	<0.01 *	−0.2 (−1.1, 0.7)	0.66	−0.5 (−1.8, 0.7)	0.40	−0.8 (−2.1, 0.6)	0.27	−1.3 (−2.6, 0.0)	0.04 *
Surgery-related side effect: self-reported food intake less than usual (6 all the time, 1 never)	−2.2 (−3.4, −0.9)	<0.01 *	−2.1 (−3.45, −0.93)	<0.01 *	−2.6 (−3.7, −1.4)	<0.01 *	0.1 (−1.6, 1.7)	0.95	0.4 (−1.4, 2.1)	0.68	0.4 (−1.2, 2.1)	0.61
Surgery-related side effect: Poor appetite (6 all the time, 1 never)	−1.2 (−2.4, −0.0)	0.04 *	−1.0 (−2.1, 0.0)	0.06	−2.2 (−3.3, −1.1)	<0.01 *	0.2 (−1.4, 1.8)	0.80	1.0 (−0.6, 2.6)	0.22	1.2 (−0.3, 2.8)	0.12
Surgery-related side effect: nausea (6 all the time, 1 never)	−0.5 (−1.7, 0.6)	0.35	−0.3 (−1.4. 0.7)	0.51	−1.6 (−2.7, −0.5)	<0.01 *	0.2 (−1.3, 1.7)	0.80	1.0 (−1.3, 1.7)	0.20	1.2 (−0.3, 2.7)	0.11
Surgery-related side effect: use of painkillers (6 all the time, 1 never)	−1.6 (−2.6, −0.6)	<0.01 *	−2.29 (−3.2, −1.4)	<0.01 *	−2.0 (−3.0, −1.1)	<0.01 *	−0.7 (−2.0, 0.7)	0.32	0.4 (−1.0, 1.8)	0.55	−0.3 (−1.6, 1.1)	0.71

Data presented as median (95% CI). Estimated changes and differences were obtained using linear mixed models adjusted for age (except for percentage outcomes where ANCOVA was used). * *p* < 0.05. Abbreviations: CI = confidence interval; DEMMI = de Morton Mobility Index; LBM = Lean Body Mass; QOL-SF: quality of life—short form survey. Overall, mean body weight change was 1.4 (−0.6, 3.4) kg in the EX + DI group (*p* = 0.17) and 0.5 (−1.8, 2.8) kg and 0.2 (−4.2, 4.6) kg in the DI and CON groups, respectively. The differences between groups were not significant.

**Table 3 nutrients-14-02599-t003:** Change in lean body mass and weight during the eight-week intervention period in postdischarge surgical patients with high training attendance and high ONS compliance (>75%).

Within Group				Between Groups					
	Exercise + Diet (>75% Training Attendance	Diet (>75% ONS Compliance)	Exercise + Diet (>75% ONS Compliance)	Exercise + Diet (>75% ONS Compliance) vs. Diet	*p*	Diet (>75% ONS Compliance) vs. Control	*p*	Exercise + Diet (>75% ONS Compliance) vs. Control	*p*
**LBM (kg)**	1.7(−0.5, 3.9)	2.1 (−0.4, 5.6)	1.9 (0.0, 3.7) *	−0.2 (−4.2, 3.7)	0.92	2.0 (−1.8, 5.8)	0.30	1.8 (−0.6, 4.2)	0.14
**LBM (%)**	3.2(−1.7, 8.0)	4.1 (−3.3, 11.5)	3.7 (−0.3, 7.6)	−1.5 (−10.1, 7.0)	0.71	3.1 (−5.0, 11.2)	0.42	1.6 (−4.4, 7.6)	0.58
**Weight (kg)**	2.9 (0.3, 5.4) *	1.4 (−4.2, 6.9)	2.0 (−0.9, 5.0)	0.7 (−5.6, 7.0)	0.83	0.9 (−5.1, 6,9)	0.76	1.6 (−2.1, 5.4)	0.39
**Weight (%)**	0.5(−2.8, 3.8)	1.4 (−7.2, 10.0)	3.1 (−1.5, 7.7)	−0.2 (−9.6, 9.2)	0.97	−0.7 (−9.5, 8.2)	0.88	−0.8 (−7.4, 5.7)	0.79

Estimated changes and differences were obtained using linear mixed models adjusted for age (except for percentage outcomes where ANCOVA was used). * *p* < 0.05. Abbreviations: LBM: lean body mass; ONS: oral nutritional supplement.

**Table 4 nutrients-14-02599-t004:** Oral Nutritional Supplement (ONS) compliance in postdischarge surgical patients.

	Diet (*n* = 13)	Exercise + Diet (*n* = 16)
**ONS compliance (%)**	50.2 [26.6, 71.9]	69.6 [40.8, 88.5]
**ONS per day (cans)**	1.0 [0.5, 1.4]	1.4 [0.8, 1.8]
**ONS compliance > 75%**		
Yes	2 (15.4%)	7 (43.8%)
No	10 (76.9%)	9 (56.2%)
**Training attendance (%)**	*NA*	98.8 [67.3, 100.0]

Data shown as median (IQR), IQR = interquartile range, ONS = oral nutritional supplements.

## Data Availability

The dataset for this study is available from the first author on reasonable request.

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
