# Peer review of "Effects of Postdischarge High-Protein Oral Nutritional Supplements and Resistance Training in Malnourished Surgical Patients: A Pilot Randomized Controlled Trial"

_nutrients, 2022, doi:10.3390/nu14132599_

Round 1
Reviewer 1 Report
Dear Authors,
The paper comes out more stronger now. However, few points remain to improve the paper.
· Line 94-97: It is still unclear what are the benefits of this design? Clear rationale why authors used this design in particular?
· Discussion (Line 320-351): I haven’t seen any changes made. Please refer to my previous comments. This section is very short and devoid of references to literature. Authors need to place the findings in a broad context.
Author Response
Thank you for your excellent comments. We have made an effort to make the suggested changes in the manuscript.
Reviewer 1:
Dear Authors,
The paper comes out more stronger now. However, few points remain to improve the paper.
Line 94-97: It is still unclear what are the benefits of this design? Clear rationale why authors used this design in particular?
Response: We agree. We have added text to the discussion section regarding why we chose this design.
Discussion (Line 320-351): I haven’t seen any changes made. Please refer to my previous comments. This section is very short and devoid of references to literature. Authors need to place the findings in a broad context.
Response: We agree. We have changed most of the discussion paragraphs and added new references.
Reviewer 2 Report
Thank you for the opportunity to review this interesting manuscript.
Materials and Methods the authors state that the patients were (line 94) “consecutively enrolled in the study between April 2018 and January 2020 at two surgical wards” That’s a rather long time to only include 45 patients. How many patients declined participation? Or was it only 57 eligible patients, I can understand Covid-19 led to problems but one is still wondering?
To address this better I think you should move lines 221-230 including Figure 1. From Results to Materials and Methods.
The heading should explain which patients the Table or Figures refer to the reader must be able to read the table without having to read the text in the manuscript.
Figure 1 should be stated what kind of patients in the figure text “post-discharge surgical patients or post-surgical patients.”
Table 1 the table text needs added information on which patients are included. All shortages in the text in the table and in the text underneath need to be explained.
Table 2 please see comments regarding Table 1. Ëšp<0.1. what is the interest to illustrate a non-significant value with Ëš. I think the Ëš should be removed from the table. One could call it fishing for positive results.
In Figure 2 please see comments regarding Figure 1 and Table 1.
In Table 3 something has happened to the text in the table so it is not readable please correct the text and see my comments regarding Table 1. Regarding Ëšp<0. Please see my comment regarding Table 2.
Table 4 please see my comments regarding Table 1.
Line 247. “The differences between groups were not significant (p>0.05)”. Suggest you remove (p>0.05) it doesn’t add anything to the text.
Line 296 The change in EX+DI was not significantly different from the changes seen for DI or CON (both p>0.05) (Table 3). Suggest you remove "(p>0.05) it doesn’t add anything to the text.
Please use the abbreviation you have chosen consequently e.g. sometimes you write EX group, Ex+Di, EX+DI, or Exercise it is the same regarding Di and Con you name them differently.
The discussion is a bit weak.
Line 332 “We found no significant differences in weight change between groups although the EX+DI group experienced the greatest weight change compared to DI and CON. However, we observed that the pattern of weight change after discharge was different between groups, as EX+DI managed to maintain their BW during intervention, whilst the other two 335 groups experienced a weight loss during the first weeks. Thus, this study shows that it is possible to achieve weight maintenance in this frail patient group when combining nutritional supplements and resistance training.” The Ex+Di group loses the most weight in total. In your conclusion, you need to address this more. I think you need to discuss this further does it matter that you lose weight if you regain it?
You further need to address the good benefits of exercise.
Why use DXA when it is much easier and better for the patients and more cost-effective to use BIA?
Author Response
Thank you for your excellent comments. We have made an effort to make the suggested changes in the manuscript.
Reviewer 2:
Thank you for the opportunity to review this interesting manuscript.
Materials and Methods the authors state that the patients were (line 94) “consecutively enrolled in the study between April 2018 and January 2020 at two surgical wards” That’s a rather long time to only include 45 patients. How many patients declined participation? Or was it only 57 eligible patients, I can understand Covid-19 led to problems but one is still wondering? To address this better I think you should move lines 221-230 including Figure 1. From Results to Materials and Methods.
Response: We agree. We have added new text regarding this to the discussion section.
The heading should explain which patients the Table or Figures refer to the reader must be able to read the table without having to read the text in the manuscript.
Response: We agree. Post-discharge surgical has been added accordingly.
Figure 1 should be stated what kind of patients in the figure text “post-discharge surgical patients or post-surgical patients.”
Response: We agree. Post-discharge surgical has been added accordingly.
Table 1 the table text needs added information on which patients are included. All shortages in the text in the table and in the text underneath need to be explained.
Response: We agree. Post-discharge surgical has been added accordingly and abbreviations used in the tables have been added throughout in the document.
Table 2 please see comments regarding Table 1. Ëšp<0.1. what is the interest to illustrate a non-significant value with Ëš. I think the Ëš should be removed from the table. One could call it fishing for positive results.
Response: We agree. The Ëš has been removed from the table.
In Figure 2 please see comments regarding Figure 1 and Table 1.
Response: We agree. Post-discharge surgical has been added in Figure 2.
In Table 3 something has happened to the text in the table so it is not readable please correct the text and see my comments regarding Table 1. Regarding Ëšp<0. Please see my comment regarding Table 2.
Response: We agree. A new table has been implemented and Ëšp<0 has been removed.
Table 4 please see my comments regarding Table 1.
Response: We agree. Post-discharge surgical has been added in Figure 2.
Line 247. “The differences between groups were not significant (p>0.05)”. Suggest you remove (p>0.05) it doesn’t add anything to the text.
Response: We agree. (p<0.05) has been removed.
Line 296 The change in EX+DI was not significantly different from the changes seen for DI or CON (both p>0.05) (Table 3). Suggest you remove "(p>0.05) it doesn’t add anything to the text.
Response: We agree. (p<0.05) has been removed.
Please use the abbreviation you have chosen consequently e.g. sometimes you write EX group, Ex+Di, EX+DI, or Exercise it is the same regarding Di and Con you name them differently.
Response: We agree. The wording has been aligned and corrected throughout the document.
The discussion is a bit weak.
Response: We agree. We have changed most of the discussion paragraphs and added new references.
Line 332 “We found no significant differences in weight change between groups although the EX+DI group experienced the greatest weight change compared to DI and CON. However, we observed that the pattern of weight change after discharge was different between groups, as EX+DI managed to maintain their BW during intervention, whilst the other two 335 groups experienced a weight loss during the first weeks. Thus, this study shows that it is possible to achieve weight maintenance in this frail patient group when combining nutritional supplements and resistance training.” The Ex+Di group loses the most weight in total. In your conclusion, you need to address this more. I think you need to discuss this further does it matter that you lose weight if you regain it?
Response: We agree. We have changed most of the discussion paragraphs and added new references.
You further need to address the good benefits of exercise.
Response: We agree. We have addressed more text on this in the discussion section.
Why use DXA when it is much easier and better for the patients and more cost-effective to use BIA?
Response: We have addressed this in the discussion section.
Round 2
Reviewer 1 Report
No further comments.
This manuscript is a resubmission of an earlier submission. The following is a list of the peer review reports and author responses from that submission.
Round 1
Reviewer 1 Report
The authors presented a study examining an interesting research topic, in which it was evaluated the acute effects of post-discharge nutritional intervention and resistance training in malnourished surgical patients. According to the authors, the results indicated a benefit from exercise and nutrition for surgical patients with malnutrition. However, only pain score and energy and protein intake differ between groups. Thus, the author's conclusion is not supported by the main outcomes. Meaningful limitations impair the study quality and may have affected the results such as the small sample size. As stated by the author, the small sample size increases the risk of committing a type II error. In addition, existing sources of bias may also have impacted the main outcomes such as adherence to resistance training, number of patients reaching 75% of their minimum energy requirements, number of patients that experienced postoperative complications. The key literature was not briefly reviewed and existing gaps or controversies in the field were not mentioned in the introduction section. Additionally, the rationale for the mentioned hypotheses was not mentioned. The resistance training program was not fully detailed (e.g., exercise types, exercise load and volume, rest interval between sets, exercise cadence, supervision rate, etc.). The discussion section only described the results. There was no attempt to explain the results.
Reviewer 2 Report
Dear Authors
The study presents interesting results but there were substantial issues raised. I recommend major revisions be made to this paper.
Specific comments
Title
Suggest: change “post-discharge nutritional intervention” to “high-protein oral nutritional supplements”.
Abstract
- Need to state the age of participants, the type of study design and the statistical analysis used.
- Line 27-28: Suggest: change diet to protein oral nutritional supplements (PONS).
Introduction
- This section is very weak in its current form. It does not provide sufficient background information on a topic. Many studies related to the topic are missing (e.g., World J Gastrointest Surg. 2016 Jul 27; 8(7): 521–532; JPEN J Parenter Enteral Nutr. Mar-Apr 2008;32(2):120-8; Surgery. 2018 Dec;164(6):1263-1270; JPEN J Parenter Enteral Nutr. 2021 Mar;45(3):596-606; J Am Coll Nutr. Sep-Oct 2020;39(7):650-656; Perioper Med (Lond). 2020 Oct 5;9:29).
- What gaps are being filled? The study needs to be a clear, scientific aim with specific hypotheses/research questions. Why this study is important in light of other studies? What this study adds?
Methods
- Line 73-82: The recruitment of participants should be clearly presented. Please also clearly describe a prospective three-arm block pilot RCT design.
- Line 101-107; Line 165-168: Please expand here. The training sessions and measuring intakes via 24-h recalls should be sufficiently described.
- Line 118-120: The use of a questionnaire should be explained in greater depth, as well as justifying their use.
Discussion
- This section is too short and devoid of studies to literature, while conclusion also need to be strengthened and expanded.
- Authors should check all their interpretations of the results. There are definitely results that are not discussed. Authors should also include hypothesis about the non-significantly differences.
- Line 314-338: meaning unclear. This section must be revised. The limitations should be described in greater details.
- There is no information as to what do these results would be helpful for future studies?